# A Positive Feedback Loop Exists between Estradiol and IL-6 and Contributes to Dermal Fibrosis

**DOI:** 10.3390/ijms25137227

**Published:** 2024-06-30

**Authors:** DeAnna Baker Frost, Alisa Savchenko, Naoko Takamura, Bethany Wolf, Roselyn Fierkens, Kimberly King, Carol Feghali-Bostwick

**Affiliations:** 1Department of Medicine, Division of Rheumatology and Immunology, Medical University of South Carolina, 96 Jonathan Lucas Street, Suite 822, MSC 637, Charleston, SC 29425, USA; bakerde@musc.edu; 2College of Osteopathic Medicine, Rocky Vista University, 4130 Rocky Vista Way, Billings, MT 59106, USA; alisa.savchenko@mt.rvu.edu; 3Department of Environmental Immuno-Dermatology, Yokohama City University Graduate School of Medicine, Yokohama 236-0004, Kanagawa, Japan; nana__26@msn.com; 4Department of Public Health Sciences, Medical University of South Carolina, 135 Cannon Street, Room 305F, Charleston, SC 29425, USA; wolfb@musc.edu; 5Barabara Davis Center, Department of Pediatrics, University of Colorado, School of Medicine, M20-3201N, 1775 Aurora Court, Aurora, CO 80045, USA; rose.fierkens@gmail.com; 6School of Medicine, Morehouse College, 720 Westview Drive, Atlanta, GA 30310, USA; kimb.king15@gmail.com

**Keywords:** dermal fibrosis, estradiol, IL-6, systemic sclerosis, aromatase

## Abstract

Systemic sclerosis (SSc) is characterized by dermal fibrosis with a female predominance, suggesting a hormonal influence. Patients with SSc have elevated interleukin (IL)-6 levels, and post-menopausal women and older men also have high estradiol (E2) levels. In the skin, IL-6 increases the enzymatic activity of aromatase, thereby amplifying the conversion of testosterone to E2. Therefore, we hypothesized that an interplay between E2 and IL-6 contributes to dermal fibrosis. We used primary dermal fibroblasts from healthy donors and patients with diffuse cutaneous (dc)SSc, and healthy donor skin tissues stimulated with recombinant IL-6 and its soluble receptor (sIL-6R) or E2. Primary human dermal fibroblasts and tissues from healthy donors stimulated with IL-6+sIL-6R produced E2, while E2-stimulated dermal tissues and fibroblasts produced IL-6. Primary dermal fibroblasts from healthy donors treated with IL-6+sIL-6R and the aromatase inhibitor anastrozole (ANA) and dcSSc fibroblasts treated with ANA produced less fibronectin (FN), type III collagen A1 (Col IIIA1), and type V collagen A1 (Col VA1). Finally, dcSSc dermal fibroblasts treated with the estrogen receptor inhibitor fulvestrant also generated less FN, Col IIIA1, and Col VA1. Our data show that IL-6 exerts its pro-fibrotic influence in human skin in part through E2 and establish a positive feedback loop between E2 and IL-6.

## 1. Introduction

Systemic sclerosis (SSc) is an autoimmune disease characterized by overproduction of extracellular matrix (ECM) components in visceral organs and skin, contributing to fibrosis [1]. In the diffuse cutaneous (dc)SSc form of the disease, progressive skin fibrosis leads to increased disability [2]. Dermal fibroblasts isolated from patients with dcSSc produce excessive IL-6, a pro-inflammatory and pro-fibrotic cytokine [3], which contributes to the accumulation of collagens and other pro-fibrotic factors in the skin [4,5]. Interestingly, severe skin fibrosis is associated with other disease manifestations, including gastrointestinal and joint involvement, muscle inflammation, restrictive lung disease, and a higher overall mortality risk [6]. These clinical observations underscore the importance of investigating the pathogenesis of skin fibrosis in SSc.

Like most autoimmune diseases, SSc has a female predominance that increases during childbearing years [7], suggesting a role for estrogen in disease pathogenesis. The most abundant form of estrogen found in non-pregnant women is estradiol (E2) [8]. E2 is pro-fibrotic in the skin and promotes the accumulation of ECM components and pro-fibrotic mediators [9,10]. Systemic E2 levels are also elevated in patients with dcSSc [9,11], with higher E2 levels associated with worse survival [11], suggesting that E2 is pathogenic in SSc.

Since no FDA-approved treatments exist for SSc-associated dermal fibrosis, the testing of potential therapies has been the focus of several clinical trials. An open-label case study of tamoxifen, an estrogen inhibitor used in breast cancer therapy, as a treatment for morphea, a form of localized scleroderma, reported regression of morphea plaques [12]. Similarly, tocilizumab, an antibody which targets the IL-6 receptor and is FDA-approved for SSc-associated interstitial lung disease, demonstrated a trend toward improved dermal fibrosis in SSc patients [13]. Therefore, IL-6 and/or estrogen signaling blockade may be viable treatment(s) for dermal fibrosis in SSc.

Aromatase (*CYP19A1*) is a cytochrome P450 enzyme active in gonadal and extra-gonadal tissues and responsible for the conversion of androgens into estrogens (i.e., testosterone to E2). It contains 10 alternative exons, a noncoding 1st exon and promoters, upstream of the 9 remaining coding exons, yielding the same protein despite alternative splicing [14]. Each noncoding exon confers tissue specificity, and the I.3 and I.4 promoters are responsible for aromatase expression in adipose and dermal tissue, respectively [15], serving as important sources for extra-gonadal aromatase [16,17]. *CYP19A1* also has binding sites upstream of the I.3 and I.4 promoters for the transcription factor and IL-6-second messenger STAT3 [15]. Thus, activated fibroblasts that produce abundant IL-6 and subsequent STAT3 activation, such as SSc dermal fibroblasts [3], can promote *CYP19A1* transcription via STAT3 [15]. This suggests that IL-6 can affect the androgen/estrogen balance in the skin through aromatase activation. The reciprocal relationship between IL-6 and E2 through aromatase activity has been established in endometrial cancer [18], but not in SSc dermal fibrosis.

In this manuscript, we explored the interaction between IL-6 and E2 using primary dermal fibroblasts from healthy donors, patients with dcSSc, and human dermal tissue in a skin organ culture model of fibrosis. We determined that E2 increased IL-6 steady-state transcript and protein levels in primary dermal fibroblasts and human dermal tissue. In turn, IL-6 increased *CYP19A1* transcript and aromatase activity levels leading to E2 production. Dermal fibroblasts treated with IL-6 also increased fibronectin (FN), collagen IIIA1 (Col IIIA1), and collagen VA1 (Col VA1) production, which was blocked by the aromatase inhibitor, anastrozole (ANA). Finally, SSc dermal fibroblasts treated with ANA or an E2 receptor signaling inhibitor, fulvestrant, decreased FN, Col IIIA1, and Col VA1 protein levels. Thus, we establish that a positive feedback loop between IL-6 and E2 exists in the skin and is exploited in IL-6-mediated dermal fibrosis, providing a potential mechanistic link between E2 and the pro-fibrotic role of IL-6.

## 2. Results

### 2.1. E2 Promotes IL-6 Steady-State mRNA and Protein Levels in Dermal Tissues and Fibroblasts

To investigate whether E2 stimulation impacts IL-6 production, we used human dermal tissues stimulated with E2 for 24 or 48 hours (h). *IL-6* steady-state mRNA levels were significantly elevated 24 h post-stimulation but returned to baseline after 48 h (Figure 1a, Appendix A). Additionally, IL-6 protein levels were significantly elevated 48 and 72 h post-stimulation (Figure 1b). Primary human dermal fibroblasts showed increased steady-state *IL-6* mRNA levels one hour compared to four hours post-E2 treatment, with a significant decline after 4 h (Figure 1c, Appendix A) and a trend toward an increase in IL-6 secreted protein levels 48 h post-E2 treatment (Figure 1d).

### 2.2. CYP19A1 Steady-State mRNA levels Increase with IL-6+sIL-6R or E2 Stimulation

Since IL-6 stimulation induces *CYP19A1* steady-state mRNA levels in endometrial cancer stromal cells using the dermal-specific promoter I.4 [18], we examined if *CYP19A1* steady-state mRNA levels increase in dermal fibroblasts in response to stimulation of E2 and/or recombinant IL-6 (IL-6) and its soluble receptor (sIL-6R). In the human skin organ culture model, we detected a statistically significant increase in *CYP19A1* steady-state mRNA levels following 24 h of E2 treatment as compared to vehicle-treated healthy donor skin tissue (Figure 2a). Primary human dermal fibroblasts produced more *CYP19A1* steady-state mRNA levels following 16 h of E2 treatment (Figure 2b). Similarly, primary human dermal fibroblasts produced more *CYP19A1* steady-state mRNA levels after 24 h of IL-6+sIL-6R stimulation (Figure 2c), but not in IL-6+sIL-6R-treated dermal tissue (Appendix A).

### 2.3. IL-6+sIL-6R Treatment Promotes Aromatase Activity and Protein in Dermal Tissue and Fibroblasts

While IL-6 treatment leads to aromatase activation in endometrial and breast cancers [18,19,20], in dermal fibroblasts, IL-6 and E2 increased *CYP19A1* transcript levels (Figure 2), suggesting increased aromatase activation. Further, positive correlations exist between increased *CYP19A1* transcript levels and localized E2 production in the skin, which occurred through aromatase activation [21]. Because E2 production can be used as a surrogate to measure aromatase activity in dermal fibroblasts [16], we evaluated if IL-6 stimulation leads to aromatase activation in dermal tissues and fibroblasts using a similar strategy. Dermal tissues and human primary dermal fibroblasts treated with IL-6+sIL-6R and testosterone, the substrate for aromatase, secreted more E2 compared to vehicle treatment (Figure 3a,b). To further verify that the secreted E2 was the result of aromatase activation, we treated dermal tissues and fibroblasts with an aromatase inhibitor, ANA, to block its activity. E2 production declined in vehicle-treated dermal tissues (Figure 3a) and fibroblasts (Figure 3b) upon ANA treatment, suggesting ANA blocks intrinsic aromatase activity. Additionally, despite IL-6+sIL-6R stimulation, ANA reduced E2 production in both dermal tissues and fibroblasts, suggesting ANA also blocks IL-6-induced aromatase activity (Figure 3a,b). We also measured aromatase total protein levels in cytoplasmic fractions from dermal fibroblasts treated with E2 and found that the protein levels were increased above the vehicle in fibroblasts from four of the five donors (Appendix A).

### 2.4. IL-6 Promotes ECM Production in Dermal Fibroblasts through Aromatase

Our data show that IL-6 promotes aromatase activity resulting in E2 production, yet it is unknown if IL-6-induced ECM is a consequence of the generated E2. Therefore, we examined how preventing IL-6-induced E2 production using ANA impacts IL-6-induced FN, Col IIIA1, and Col VA1 protein production, since both E2 and IL-6 can independently amplify FN, Col IIIA1, and Col VA1 levels [5,9,10,22]. Primary human dermal fibroblasts from healthy donors produced more Col IIIA1, Col VA1, and FN protein when treated with IL-6+sIL-6R compared to the vehicle, with a decrease in protein levels after ANA treatment (Figure 4, Appendix A). These data suggest that IL-6-induced Col IIIA1, Col VA1, and FN are mediated by E2 through aromatase activation.

### 2.5. SSc Dermal Fibroblasts Produce More Steady-State CYP19A1 Transcripts and Show Increased Responsiveness to Aromatase Inhibition

Patients with dcSSc have the following: 1. activated dermal fibroblasts that secrete high levels of IL-6 [3] and 2. high circulating IL-6 [23] and E2 [6,8] levels, both of which can increase *CYP19A1* mRNA levels ([16], Figure 2). Therefore, we measured *CYP19A1* transcript levels in dcSSc primary dermal fibroblasts derived from patients with < 1 year disease duration. Compared to primary dermal fibroblasts from healthy donors, dermal fibroblasts from patients with dcSSc expressed significantly higher steady-state *CYP19A1* transcripts (Figure 5a). Dermal fibroblasts from dcSSc patients also secreted E2 upon treatment with IL-6+sIL-6R that was blocked by ANA treatment, suggesting aromatase activation (Figure 5b). Interestingly, these fibroblasts were also more responsive to ANA treatment, significantly reducing levels of secreted E2 as compared to healthy donor dermal fibroblasts despite treatment with IL-6+sIL-6R (Figure 5c).

### 2.6. E2 Production Blockade Decreases FN, Col IIIA1, and Col VA1 Transcript and Protein Levels in dcSSc Dermal Fibroblasts

Since dcSSc dermal fibroblasts produce excessive ECM [24], we examined the effects of aromatase activity blockade on ECM production in these effector cells. SSc dermal fibroblasts treated with ANA generated lower FN, Col IIIA1, and Col VA1 transcript and protein levels compared to the vehicle (Figure 6a–c). Therefore, inhibiting E2 production in dcSSc dermal fibroblasts reduces ECM production through aromatase.

### 2.7. E2 Receptor Signaling Blockade Decreases FN and Type IIIA1 and VA1 Collagen in dcSSc Dermal Fibroblasts

Estrogen propagates its signaling through estrogen receptor α (ERα) to induce fibrosis [9,10]. Therefore, we investigated if the E2 produced by dcSSc dermal fibroblasts engages in receptor signaling to contribute to dermal fibrosis. We assessed if disrupting E2 signaling using fulvestrant (ICI), a known ERα antagonist [25], impacts FN and collagen IIIA1 and VA1 protein levels in dcSSc dermal fibroblasts. Dermal fibroblasts from dcSSc patients treated with fulvestrant produced less FN and collagen type IIIA1 and VA1 protein compared to vehicle-treated dcSSc fibroblasts (Figure 7a–c).

### 2.8. Sex-Dependent ECM Expression in Dermal Fibroblasts of Mice Genetically Deficient in ERα

Despite ERα being one of the major receptors responsible for cellular E2 signal propagation [26], classical and alternative isoforms of the receptor exist with distinctive functions [27,28,29]. Since ECM production in dcSSc fibroblasts was abrogated by ICI, suggesting that E2 signaling is via ER*α* (Graphical Abstract), we examined the role of ERα in IL-6-induced FN, Col IIIA1, and Col VA1 protein levels using primary dermal fibroblasts isolated from mice that are functionally deficient in the classical ERα (ERKO) [30]. In dermal fibroblasts from female mice, no difference was detected in Col VA1 protein production between wild-type (WT) and ERKO animals treated with IL-6+sIL-6R, but less FN and more Col IIIA1 protein were produced by dermal fibroblasts lacking the classical ERα (Appendix A). However, dermal fibroblasts isolated from WT and ERKO male mice that were treated with IL-6+sIL-6R did not display differences in FN, Col IIIA1, or Col VA1 protein levels (Appendix A). Taken together, these data suggest that dermal fibroblasts that are genetically deficient in the classical ERα have variable ECM protein expression depending on the sex of the animal, likely due to the presence of other ER*α* isoforms.

## 3. Discussion

Autoimmune diseases, such as systemic lupus erythematosus, rheumatoid arthritis, and SSc, are characterized by a microenvironment composed of inflammatory cytokines and abnormal levels of sex hormones [11,31,32,33]. There is a clear association between chronic inflammation and fibrosis mediated through IL-6 [34,35]. Few studies have examined the intersection between inflammation and sex hormones in autoimmune diseases [21,36]. In this manuscript, we propose that the relationship between IL-6 and the sex hormone E2 that was first described in the tumor microenvironment [18,37] exists in the skin and contributes to dermal fibrosis. In diseases with an inflammatory element and a sex bias, our findings suggest a possible explanation connecting the observed hormonal and inflammatory elements to chronic fibrosis.

Our data show that healthy dermal fibroblasts and tissues produced higher *CYP19A1* steady-state mRNA levels in response to IL-6+sIL-6R treatment. In turn, IL-6 activated the aromatase enzyme to catalyze the conversion of testosterone to secreted E2, intimating that IL-6 induces both *CYP19A1* transcription in dermal fibroblasts and aromatase activation in dermal fibroblasts and tissues. We did not observe an increase in *CYP19A1* transcript levels in dermal tissue, possibly due to the presence of other cell types that may not respond to IL-6+sIL-6R by producing *CYP19A1* transcripts. Because the *CYP19A1* gene promoter, which contains an upstream STAT3 binding site, is specific for fibroblasts [38,39], *CYP19A1* transcriptional differences in skin may be blunted by resident cells that do not generate *CYP19A1* in response to IL-6. Our data also confirm that IL-6+sIL-6R is pro-fibrotic in the skin and increases FN and VA1 collagen levels, with a lesser impact on type III collagen. However, the production of these proteins decreased by impeding E2 production using the aromatase inhibitor, ANA. Because these proteins can accumulate in response to E2 treatment as well [9,10,22], our data suggest that E2 may serve as a mediator for IL-6 in dermal fibroblasts through aromatase. These findings are novel because they suggest a mechanistic partnership between inflammation and fibrosis by way of E2. In an animal model of radiation-induced pulmonary fibrosis, both fibrosis and inflammation improved after treatment with aromatase inhibitors, resulting in decreased fibrosis and reduced levels of pro-fibrotic mediator transcripts and IL-6 protein [40,41], implying that aromatase blockade diminishes both fibrosis and inflammation in the lung. Taken together, these findings highlight that increased aromatase activity leads to E2 generation, and that E2 production blockade by aromatase inhibition is a promising therapeutic option.

After E2 is generated, it can, in turn, increase IL-6 transcript and protein levels in dermal tissue and fibroblasts. Several studies hypothesized the intracellular pathway used by E2 leading to IL-6 production. In endometrial cancer cells, E2 can augment IL-6 production, and it is suggested that this occurs through the NFκB pathway [18]. Similarly, through examining ovariectomized female rats that were treated with E2, it was discovered that the synovial membrane in the temporomandibular joint generates IL-6 in a dose-dependent manner, likely through the NFκB pathway [42]. Based on these studies, we hypothesize comparably that E2 signals through the NFκB pathway to increase IL-6 production in dermal fibroblasts and tissue.

Baseline *CYP19A1* transcripts are elevated in dermal fibroblasts from dcSSc patients with short disease duration (<1.0 year). Since we and others demonstrate that IL-6+sIL-6R increases *CYP19A1* [18], elevated *CYP19A1* levels detected in dcSSc fibroblasts may be the result of the known high IL-6 levels which can create autocrine feedback [3] to maintain the increased *CYP19A1* transcript levels. The resulting increase in local E2 production may then contribute to both systemic E2 levels [9,11,31] and continued *CYP19A1* transcription in conjunction with IL-6. Additionally, patients with SSc may be genetically predisposed to produce high levels of E2. Several single-nucleotide polymorphisms and genetic variants were identified within *CYP19A1* introns which influence its regulatory elements and mRNA splicing, leading to aromatase activation and elevated systemic E2 levels [43,44]. However, few studies have evaluated *CYP19A1* genetic variations in extra-gonadal tissues such as the skin [21]. Future directions include examining polymorphisms within the *CYP19A1* gene of dcSSc patients that may predispose them to exaggerated systemic E2 production and contribute to worse disease outcomes.

We show that IL-6-induced Col VA1 expression in dcSSc dermal fibroblasts is regulated through E2 production. Like E2, Col V may serve as a link between fibrosis and inflammation. Animals immunized with Col V develop both an inflammatory and a fibrotic response in the lung when compared to immunization with type I or type III collagen [45,46]. Recent data suggest that Col V may serve as a self-antigen activating and perpetuating the autoimmune response [47]. Patients with SSc have circulating anti-Col V antibodies [48] that are specific against known immunogenic portions of Col V and can bind to SSc lung tissues [49]. Therefore, diminishing IL-6-induced Col V through E2 modulation may affect both the inflammatory and fibrotic networks of IL-6.

In dcSSc dermal fibroblasts, E2 receptor signaling blockade using fulvestrant decreased IL-6 and E2 induction of FN and type IIIA1 and VA1 collagen [5,9,10,22], yielding similar results to aromatase inhibition. Healthy donor dermal fibroblasts treated with fulvestrant reduced E2-induced FN and collagen 22A1 protein production [9,10]. Other studies have corroborated the role of ERα signaling in fibrosis. Inhibition of ERα resulted in decreased dermal fibrosis in skin tissues maintained in organ culture [9], and selective estrogen receptor modulators are suggested to have an anti-fibrotic effect in SSc [50]. Therefore, our data suggest E2 produced through IL-6-induced aromatase activity actively contributes to dermal fibrosis by signaling through ERα to activate downstream pro-fibrotic mediators. We demonstrate the interconnectedness between IL-6 and E2 in the production of these pathogenic fibrotic mediators at the RNA and protein levels in a diseased state. The novelty of our findings is that they not only support that IL-6 and E2 are interrelated in the skin, but also that they function together, contributing to the pathogenesis of dermal fibrosis in the hormonal and inflammatory scleroderma microenvironment.

We found that sex impacts the IL-6-induced ECM protein production in ERα-deficient murine dermal fibroblasts. Studies show correlations between ERα and FN transcript levels in the skin [51], suggesting that E2-induced FN levels may be mediated by ERα signaling. Likewise, our data show that IL-6-induced FN protein levels are reduced when the classical ERα protein is deficient in ERKO female mice and upon treatment of dcSSc dermal fibroblasts with fulvestrant. However, differing results were obtained when measuring IL-6-induced Col IIIA1 and Col VA1 protein in female ERKO vs. WT dermal fibroblasts compared to dcSSc dermal fibroblasts treated with fulvestrant. Although ERKO mice lack the classical ERα isoform, they express alternative ERα isoforms which have tissue specificity and differing sex-specific functions [27,29] distinct from those of the classical ERα. Additionally, fulvestrant treatment impacts the availability of ERα isoforms [52]. Thus, the dissimilarities in the response of fibroblasts following ERα chemical inhibition and ERα genetic deficiency is likely due, in part, to the presence of ERα isoforms that retain activity in ERKO mice. Future directions include examining the specific role of the different ERα isoforms in dermal fibrosis.

Clinical trials using estrogen-based hormone replacement therapies in healthy volunteers documented increased skin thickness in the participants [53,54,55]. We show that E2 increased levels of FN and collagens IIIA1 and VA1. Likewise, we previously reported that E2 increases collagen, *FN*, and *TGFβ1* levels [9,10]. Here, we propose that E2 produced through aromatase not only directly influences ECM components in the skin, but also indirectly increases the levels of ECM through the pro-fibrotic cytokine IL-6 based on the positive feedback loop between E2 and IL-6. While tocilizumab is FDA-approved for treatment of SSc-associated ILD through targeting IL-6 signaling, our results implicate E2 and IL-6 as contributors to dermal fibrosis in dcSSc and suggest that effective therapies for dcSSc and other autoimmune diseases likely require concomitant inhibition of both E2 and IL-6.

## 4. Materials and Methods

### 4.1. Human Skin Organ Culture Ex Vivo

Discarded skin samples from healthy donors who underwent skin-resection procedures were used as previously described [9,56,57,58]. The demographic information of the donors is summarized in Table 1. Six 3 mm punches/well were placed dermal side down in 6-well tissue culture dishes (Costar, Corning, NY, USA) and maintained in serum-free, phenol-red-free DMEM (Corning, Corning, NY, USA). Tissues were stimulated with a vehicle (ETOH (Hyclone, South Logan, UT, USA)), 1X PBS (Corning, Corning, NY, USA)), E2 (10 nM, Tocris, Minneapolis, MN, USA), or recombinant human IL-6 + recombinant human soluble IL-6 receptor (IL-6+sIL-6R, R and D Systems, Minneapolis, MN, USA), both at 20 ng/mL. In all experiments, skin punches and supernatants were harvested at the specified times and stored at −80 °C until further evaluation.

### 4.2. Primary Human Dermal Fibroblast Culture

Primary human dermal fibroblasts were cultured using the outgrowth method [3] from skin tissue isolated from the abdomen, forearm, upper arm, or shoulder of healthy donors or patients with dcSSc under a protocol approved by the IRB of the Medical University of South Carolina. The demographics and disease duration of the dcSSc patients are summarized in Table 1. Fibroblasts were used between passages 3 and 8 and were plated at 1.5–2.0 × 10^5^ cells/well in 6-well culture dishes (Costar, Corning, NY, USA). Primary dermal fibroblasts from healthy donors were treated with E2 or IL-6+sIL-6R at the same concentrations as human skin above or their respective vehicles (ETOH or PBS, respectively) in serum-free, phenol-red-free DMEM for the indicated time points. In some experiments, primary human dermal fibroblasts from dcSSc patients were treated with PBS or IL-6+sIL-6R.

### 4.3. Measurement of Steady-State mRNA Levels

Total RNA was isolated from human skin tissue and primary fibroblasts using TRIzol (Invitrogen, Carlsbad, CA, USA). Steady-state mRNA levels were measured using quantitative PCR (qPCR) and calculated using the delta–delta CT method, represented as the fold change over vehicle following normalization of the signal to *B2M* (Hs00187842_m1) or *GAPDH* (Hs02758991_g1). Baseline or vehicle-treated steady-state mRNA levels were calculated using the delta CT method, represented as the relative quantity following normalization of the signal to *B2M* or *GAPDH*. Primers specific for *CYP19A1 *(Hs00903411_m1), *FN* Hs00365052_m1), *Col IIIA1* (Hs00943809_m1), *Col VA1* (Hs00609133_m1), and *IL-6* (Hs00985639_m1) as well as *B2M * and *GAPDH* were purchased from Thermofisher (Rockford, IL, USA).

### 4.4. IL-6 ELISA

Human skin tissue in organ culture and primary human dermal fibroblasts were stimulated with E2 or a vehicle (ETOH) for the designated time points. The supernatants were harvested, and secreted IL-6 protein was measured using a human IL-6 ELISA (Enzo Life Sciences, Farmingdale, NY, USA). The data represent the relative quantity of measured protein in relation to each vehicle-treated sample.

### 4.5. Aromatase Activity Assay

Ex vivo human skin tissue or primary dermal fibroblasts were stimulated with IL-6+sIL-6R for 24 or 48 h, respectively, and 10 nM of testosterone (Pfizer, New York, NY, USA) was added as the substrate for aromatase for an additional 48 h. ANA (dissolved in 100% DMSO, Accord, Durham, NC, USA) was added at the same time as testosterone. E2 production was used as a surrogate for aromatase activity. E2 was measured in undiluted supernatant samples in duplicate using ELISA (Calbiotech, Spring Valley, CA, USA). The data represent the relative quantity of measured protein in relation to each vehicle-treated sample.

### 4.6. Subcellular Fractionation

Primary human dermal fibroblasts were plated in a 10 cm culture dish and treated with ETOH or E2 [10 nM] for 48 h. At the time of harvest, the cytoplasmic fractions were isolated from the dermal fibroblasts using a subcellular protein fractionation kit (ThermoFisher Scientific, Waltham, MA, USA) with the procedures followed as outlined in the provided protocol.

### 4.7. Immunoblot Analysis

Whole-cell protein lysates (harvested in RIPA buffer) or cytoplasmic fractions from primary human dermal fibroblasts were resolved using 10% SDS PAGE and transferred to a nitrocellulose membrane (GE Healthcare Life Sciences, Pittsburgh, PA, USA). Following blocking with 5% non-fat dry milk, membranes were incubated with the following primary antibodies: FN (clone EP5, Santa Cruz Biotechnology, Santa Cruz, CA, USA), FN (clone F14, Abcam, Waltham, MA, USA), alpha tubulin (clone DM1A, Abcam, Waltham, MA, USA), Col IIIA1 (Proteintech, Rosemont, IL, USA), Col VA1 (clone C-5, Santa Cruz Biotechnology, Santa Cruz, CA, USA), aromatase (Invitrogen, Waltham, MA, USA), and GAPDH (clone 0411, Santa Cruz Biotechnology, Santa Cruz, CA, USA). This was followed by incubation with horseradish peroxidase-conjugated secondary antibodies: goat anti-mouse IgG (Promega, Madison, WI, USA) and donkey anti-rabbit IgG (Cytiva, Malborough, MA, USA). After washing, immunoblots were developed with chemiluminescence reagents according to the manufacturer’s protocol (Pierce, Rockford, IL, USA). Signal intensities were quantified by densitometry, normalized against GAPDH or alpha tubulin in each sample, and expressed as the magnitude of increase compared with controls. Densitometry was calculated using Image J v.1.53t [59].

### 4.8. Statistical Analysis

All data were tested for normal distribution. Non-normally distributed data were log-transformed to obtain normality. If the data were normally distributed, a two-way ANOVA, one-way ANOVA with post hoc test analysis, or paired or unpaired two-sample *t*-test was used to determine statistical significance, which was defined as a *p*-value < 0.05 using GraphPad (v. 10).

## Figures and Tables

**Figure 1 ijms-25-07227-f001:**
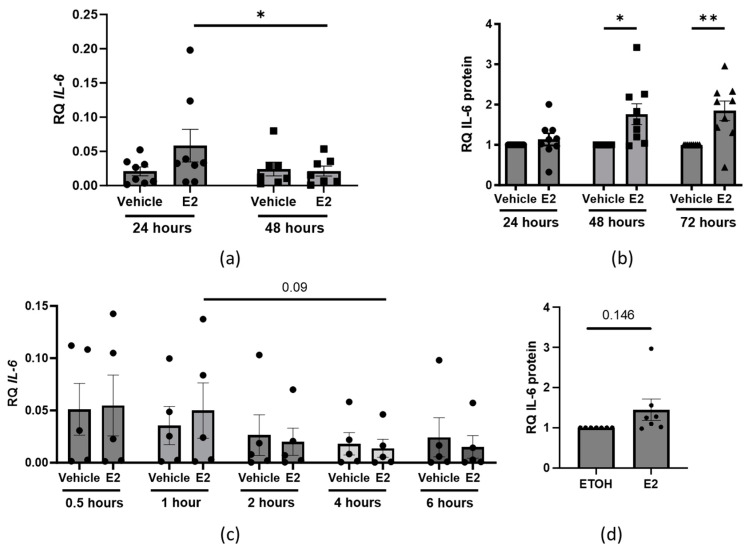
Relative quantity (RQ) IL-6 steady-state transcript and protein levels in E2-treated dermal tissue and fibroblasts. (**a**) RQ *IL-6* steady-state mRNA 24 or 48 h post-vehicle (ETOH) or E2 treatment [10 nM] in human skin tissue ex vivo. *n* = 7–8 individual samples, two-way ANOVA. (**b**) RQ of secreted IL-6 protein 24, 48, or 72 h post-ETOH vs. E2 treatment in skin tissue ex vivo. *n* = 9 individual samples. (**c**) RQ steady-state mRNA levels of *IL-6* post-vehicle vs. E2 treatment in human dermal fibroblasts from healthy donors. (**d**). RQ of secreted IL-6 protein 48 h post-ETOH vs. E2 treatment in human dermal fibroblasts in vitro. *n* = 5 individual cell lines, two-way ANOVA with post−hoc analysis (**a–c**) after log transformation (**c**) or paired *t*-test (**d**). Error bars = SEM. * *p* < 0.05, ** *p* < 0.01.

**Figure 2 ijms-25-07227-f002:**
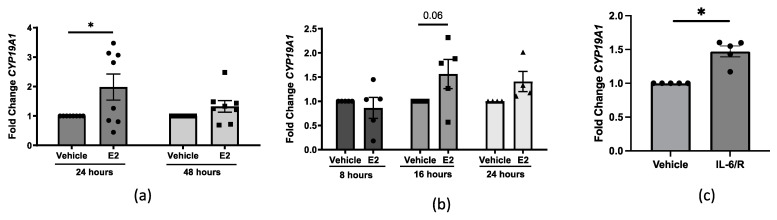
Steady-state *CYP19A1* transcript levels in dermal tissue ex vivo and fibroblasts in vitro. (**a**) Steady-state *CYP19A1* mRNA levels 24 and 48 h post-vehicle vs. E2 treatment in healthy donor human skin tissue ex vivo. *n* = 8 individual samples, two-way ANOVA. (**b**) Steady-state *CYP19A1* mRNA levels 8, 16, or 24 h post-vehicle vs. E2 treatment [10 nM] in human primary dermal fibroblasts. *n* = 4–5 individual cell lines, two-way ANOVA. (**c**) Steady-state mRNA levels *CYP19A1* 24 h post-vehicle (phosphate-buffered saline (PBS)) vs. IL-6+sIL-6R [20 ng/mL] treatment in primary human dermal fibroblasts. *n* = 5 individual cell lines, paired *t*-test. Error bars = SEM. * *p* < 0.05.

**Figure 3 ijms-25-07227-f003:**
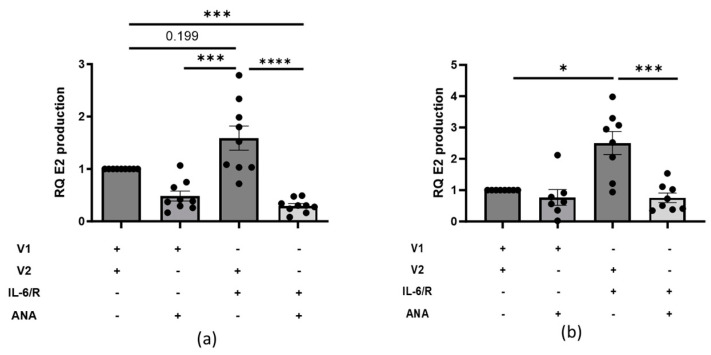
IL-6-induced aromatase activity in dermal tissue ex vivo and dermal fibroblasts in vitro. (**a**) Relative quantity (RQ) of secreted E2 production 72 h post-IL-6+sIL-6R [20 ng/mL] treatment in human skin tissue ex vivo. *n* = 9 individual samples, one-way ANOVA after log transformation with post−hoc analysis. (**b**) RQ of secreted E2 production 96 h post-IL-6+sIL-6R [20 ng/mL] treatment in human primary dermal fibroblasts in vitro. *n* = 5 individual cell lines with 3 measured in duplicate, one-way ANOVA after log transformation with post−hoc analysis. In all conditions, 10 nM of testosterone was added to both skin tissue and fibroblast cultures. V1 = PBS, V2 = DMSO. Error bars = SEM. * *p* < 0.05, *** *p* < 0.005, **** *p* < 0.0001.

**Figure 4 ijms-25-07227-f004:**
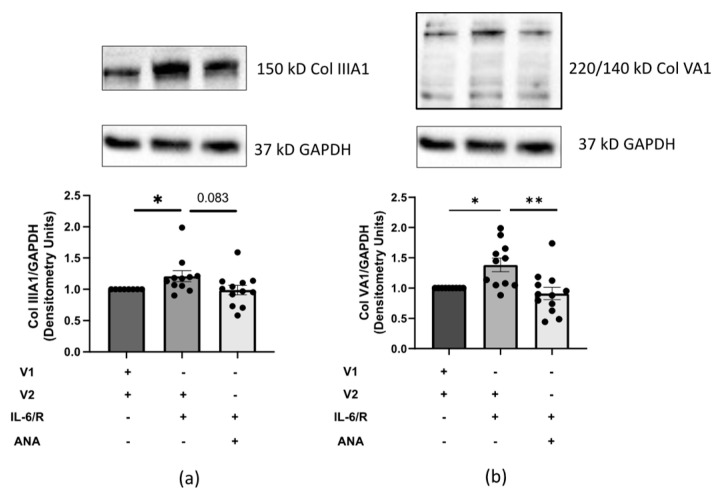
IL-6-induced type III Col A1 and type V Col A1 protein levels in dermal fibroblasts. Representative immunoblots and densitometry of Col IIIA1 (**a**) and Col VA1 (**b**) protein levels in primary human dermal fibroblasts from healthy donors treated for 96 h with IL-6+sIL-6R [20 ng/mL]. Fibroblasts were treated with DMSO or ANA [300 nM] 1 h prior to PBS or IL-6+sIL-6R. For Col VA1 (**b**), both the 220 and 140 kD bands were used for calculation. Protein levels were normalized to GAPDH and presented as fold change. V1 = PBS, V2 = DMSO. *n* = 4 individual cell lines used in 9 independent experiments. One-way ANOVA with post hoc analysis. Error bars = SEM. * *p* < 0.05, ** *p* < 0.01.

**Figure 5 ijms-25-07227-f005:**
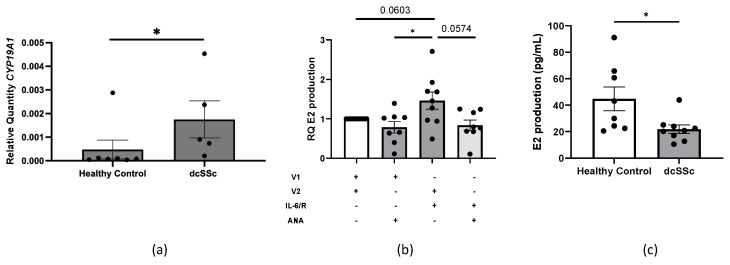
*CYP19A1* relative quantity and aromatase enzymatic activity in dcSSc dermal fibroblasts in vitro. (**a**) Steady-state mRNA levels of *CYP19A1* at baseline in primary dermal fibroblasts from healthy donors vs. dcSSc patients. *CYP19A1* gene expression was normalized to *beta 2 micoglobulin* (*B2M*). *n* = 6 individual healthy donor cell lines (one healthy control donor measured in duplicate), *n* = 5 SSc donor fibroblast cell lines, *p*-value calculated using unpaired *t*-test after log transformation. (**b**) Relative quantity (RQ) of secreted E2 produced 96 h post-IL-6+sIL-6R [20 ng/mL] treatment in SSc primary dermal fibroblasts with vehicle or ANA inhibition [300 nM]. *n* = 6 individual cell lines, 4 measured in duplicate, one-way ANOVA with post hoc analysis. (**c**) Comparison of E2 production in primary dermal fibroblasts from healthy donors vs. dcSSc patients post-IL-6+sIL-6R [20 ng/mL] treatment and ANA inhibition [300 nM]. Data are relative to fibroblasts treated with their respective vehicles. *n* = 5 individual cell lines, measured in duplicate. Unpaired *t*-test. Error bars = SEM. * *p* < 0.05.

**Figure 6 ijms-25-07227-f006:**
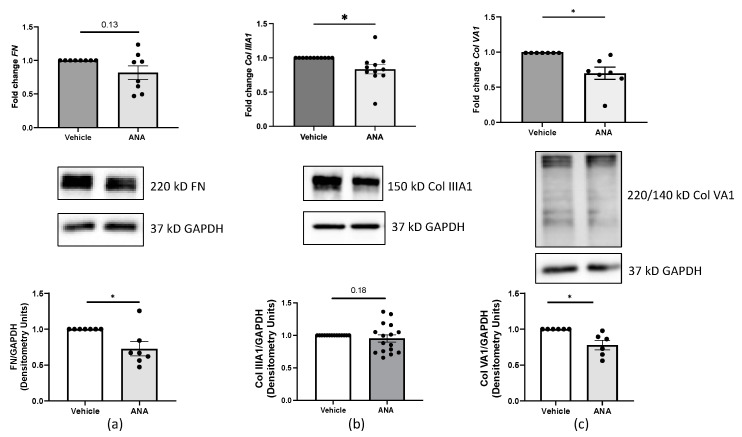
Steady-state ECM transcript and protein levels in ANA-treated dcSSc dermal fibroblasts. *FN* (**top a**), *Col IIIA1* (**top b**), and *Col VA1* (**top c**) fold change transcript levels were quantified in primary dermal fibroblasts from dcSSc patients 48 h (**a,c**) or 96 h (**b**) post-treatment with ANA [300 nM]. mRNA levels were normalized to *GAPDH*. Representative immunoblots and densitometry of FN (**bottom a**), Col IIIA1 (**bottom b**), and Col VA1 (**bottom c**) protein levels in primary dermal fibroblasts from dcSSc patients 120 h (**a**,**c**) or 96 h (**b**) post-treatment with ANA [300 nM]. For Col VA1 (**c**), both the 220 and 140 kD bands were used in the calculation. Protein levels were normalized to GAPDH and presented as fold change. *n* = 5 individual cell lines used in 6–14 independent experiments. Error bars = SEM. Paired *t*-test, protein data log transformed, * *p* < 0.05.

**Figure 7 ijms-25-07227-f007:**
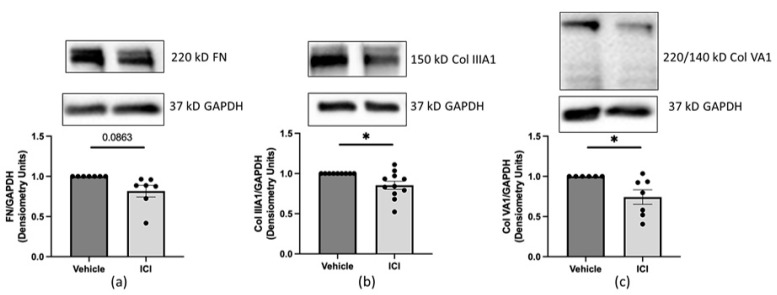
Reduction in FN and type IIIA1 and VA1 collagen protein levels produced by dcSSc dermal fibroblasts after fulvestrant treatment. Representative immunoblots and densitometry of FN and type IIIA1 and VA1 collagen protein levels in primary dermal fibroblasts from dcSSc patients 96 (**a**,**c**) or 120 (**b**) hours post-treatment with fulvestrant (ICI) [100 nM]. Protein levels were normalized to GAPDH. For Col VA1 (**c**), both the 220 and 140 kD bands were used in the calculation. *n* = 7 individual cell lines in 7 (**a**,**c**) and 11 (**b**) independent experiments. Error bars = SEM. Paired *t*-test, * *p* < 0.05.

**Table 1 ijms-25-07227-t001:** Demographics of cell and tissue donors.

			dcSSc Fibroblasts In Vitro
ID	Age at Biopsy (Years)	Sex	Disease Duration (Years)
SSc 001	54	M	2.5
SSc 002	30	F	4
SSc 003	69	M	3
SSc 005	42	M	5
SSc 006	69	F	2.5
SSc 007	46	M	3
SSc 120	45	F	<1
SSc 121	47	F	<1
SSc 122	43	F	<1
SSc 102	38	F	<1
SSc 103	67	F	<1
**Healthy donor fibroblasts**			
**ID**	**Age (years)**	**Sex**	
6	47	F	
13	54	F	
22	45	F	
24	60	F	
48	48	F	
60	41	F	
61	61	F	
64	71	F	
101	unknown	unknown	
102	unknown	unknown	
115	40	F	
118	unknown	unknown	
139	unknown	unknown	
172	26	F	
174	44	F	
175	66	F	
177	34	F	
180	48	F	
210	61	F	
216	34	F	
228	50	F	
229	39	F	
230	37	F	
237	38	F	
238	42	F	
256	unknown	F	
315	29	F	
331	37	F	
333	42	F	
343	35	F	
348	56	F	
351	29	F	
**Dermal tissue ex vivo**			
**ID**	**Age (years)**	**Sex**	
4	36	F	
10	25	F	
14	49	F	
17	39	F	
20	34	F	
33	35	F	
188	44	F	
197	53	F	
198	65	F	
216	34	F	
217	46	F	
220	36	F	
227	51	F	
229	39	F	

Female (F), male (M).

## Data Availability

The original data presented in the study are openly available in FigShare at DOI:10.6084/m9.figshare.25939741.

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
