# Peer review of "A Positive Feedback Loop Exists between Estradiol and IL-6 and Contributes to Dermal Fibrosis"

_ijms, 2024, doi:10.3390/ijms25137227_

Round 1

Reviewer 1 Report

Comments and Suggestions for Authors

Thanks for providing this manuscript for peer-reviewing.

In this study, the authors investigated the interplay between the estradiol (E2) and IL-6 and their contribution to the dermal fibrosis in diffuse cutaneous (dc) Systemic Sclerosis (SSc).  The study suggests that IL-6 enhances the production of E2 in the skin by increasing aromatase activity, which in turn augments fibrotic processes in this disease. Subsequently the authors are proposing a potential therapeutic strategy for managing dermal fibrosis in SSc by disrupting the positive feedback loop between E2 and IL-6.

While the research is impactful, and this work could be published in this journal. I have some major comments that need to be addressed before to acceptance and publication of the manuscript.

1-      Please provide all IL6 analysis in figure 1 comparing it to the vehicle controls ; the fold changes doesn’t reflect a biological significance here. Please consider re-evaluating the statical analysis or providing additional data compared to the vehicle to support your conclusion.

2-      Figure 2 does IL6/R increase CYP19A1  in human skin as in E2 “A”?  Please provide similar results to A/B with IL6 to conclude that IL6/R increase CYP19A1

3-      Please provide protein levels OF CYP19A1 as increase mRNA only doesn’t necessary indicate increased aromatase activity.

4-      IL6 in Figure 3 A, does not increase E2 compared to control. The observed significance may be driven primarily by ANA. This does not support the conclusion that Il6 increase aromatase activity.

5-      Fig 3, please confirm whether testosterone was added to all conditions or provide a testosterone-only control.

6-      The conclusion that IL-6 induces ColIIIA1 through aromatase activation may be inconclusive, as it relies on only 2 out of 9 responses. Consider discussing the limitations and implications of this finding.

7-      Please compare Figure 5B to the controls to highlight any significant differences.

8-      Please ensure that Supplementary Figures are in an appropriate format for clarity and readability. Providing them in way of PPT is so confusing.

Comments on the Quality of English Language

Writing needs to be improved for clarity and all the material and method section need more details (such as materials lots or catalogue numbers, dilutions of Antibodies used for immunoblotting etc..)

Reviewer 2 Report

Comments and Suggestions for Authors

This is an interesting paper examining the interplay between E2 and IL6, in the context of aromatase. The study design is pretty straight forward and easy to follow. Data analysis is sound. The authors used a robust number of samples for the assays, which is a major strength. There are a few areas that still need improvement and clarification.

(1) The supplemental figures are missing. The reviewer is excited about the mouse data but none of it can be found in this submission. 

(2) Can the authors speculate or provide additional data on on E2 stimulates IL6 levels? 

(3) For the Western blot for COLVA1, multiple bands are present. How did the authors quantify the bends?

(4) In Figure 6, the label (a) is missing from the figure.

Round 2

Reviewer 1 Report

Comments and Suggestions for Authors

Thanks for updating the manuscript and addressing my previous comments. Good Luck.